# OpenReview forum: "ChartMaster: Boosting MLLMs for Chart Analysis through Data, Perception, and Reasoning Optimization"
_ICLR.cc/2026/Conference — Submitted to ICLR 2026_

### Official Review · Reviewer_e9y4 · 2025-10-27

**Soundness:** 3
**Presentation:** 2
**Contribution:** 2
**Rating:** 4
**Confidence:** 4

**Summary:**

This paper introduces ChartMaster, a framework to boost MLLMs for chart analysis through optimizations in data, perception, and reasoning. Key components include: (1) ChartVerse, a 128k-instance synthetic dataset with diverse charts and QA pairs via a decoupled generation pipeline (95% error-free); (2) Multi-Negative Direct Preference Optimization (MNDPO), extending DPO with hard negatives for precise visual extraction; (3) Reinforcement Learning with Dynamic Length Reward (DLR) for adaptive reasoning depth. Trained in two stages, it achieves SOTA on six benchmarks, rivaling proprietary models.

**Strengths:**

The construction of ChartVerse is a contribution, providing a high-quality, diverse dataset (128k instances) with an LLM code-based synthesis pipeline that ensures better alignment between charts and QA pairs.

The holistic framework addresses key challenges in chart analysis (perception and reasoning) in a structured way, leading to consistent improvements across benchmarks in the main results (Table 1).

The ablation studies provide some insight into hyperparameters, such as the number of negatives in MNDPO (N=1 to 5) and the insensitivity of DLR to its temperature parameter τ.

**Weaknesses:**

The proposed MNDPO lacks significant novelty, as similar multi-negative extensions of Direct Preference Optimization have been explored in recent works[1], which also leverage multiple negatives for multimodal alignment. This makes MNDPO feel like a minor variant on multi-modal chart understanding tasks.

The Dynamic Length Reward (DLR) mechanism in the RL stage appears to have negligible practical impact. Ablations show it achieves the "best overall performance," but the gains are minimal or absent in terms of accuracy (no performance drop without it, per the text), and it primarily reduces average output length by only around 20 tokens across benchmarks. This reduction is not substantial enough to meaningfully improve efficiency or mitigate overthinking in real-world scenarios, especially given the variable nature of chart queries.

Closed-source models compared are not the latest, as current closed-source models come to GPT-5 or Claude-4.5, although I get that beating these models is not the goal.

Overall, the paper feels incremental rather than transformative. The core ideas rely heavily on established techniques (synthetic data generation, DPO variants, and RL for reasoning), with limited evidence of superior generalization or robustness beyond the reported benchmarks. The claims of rivaling proprietary models are strong but not deeply substantiated against the latest closed-source systems, and the work could benefit from more rigorous comparisons or analyses of failure cases.

[1]Chen et al. On Softmax Direct Preference Optimization for Recommendation. NeurIPS 2025

**Questions:**

See Cons

In line 356, I think ChartMaster is built upon Qwen2.5-7B-VL, not the Qwen2.5-7B-Instruct model, right? I suppose Qwen2.5-7B-Instruct is more likely referred to as the text model.

---

> ### Author Response · Authors · 2025-11-21
> **Response to Reviewer e9y4 (Part 1 / 2)**
>
> We sincerely thank you for recognizing the contributions of ChartVerse and our holistic framework. We appreciate your constructive criticism regarding the novelty of MNDPO and the impact of DLR. Below, we address your concerns point-by-point to clarify the distinctiveness and significance of our contributions.
>
> # Is MNDPO a minor variant for multimodal chart understanding tasks?
>
> We respectfully disagree with the characterization of MNDPO as a minor variant.
> While we acknowledge that multi-negative strategies exist in other domains (e.g., recommendation systems), MNDPO is fundamentally distinct from works like S-DPO [1] in formulation, domain application, and sample construction:
>
> - Distinct Mathematical Formulation: Just as IPO[2], KTO[3], and SimPO[4] are distinct advancements rather than "minor variants" of DPO, MNDPO utilizes a specific  competition objective tailored for fine-grained discrimination, which is mathematically distinct from the ranking losses used in S-DPO [1].
> - Domain-Specific Design: S-DPO is designed for text-based recommendation systems. In contrast, MNDPO is the first framework specifically tailored for multimodal chart perception. The challenges are vastly different: recommendation focuses on semantic relevance, while chart analysis requires precise visual acuity (e.g., distinguishing a bar height of 35.2 from 35.3).
> - Hard Negative Construction: S-DPO typically uses "sub-optimal candidates" (retrieved negatives). Conversely, we actively synthesize hard negatives (lausible but incorrect answers) with subtle visual or numerical perturbations (e.g., adjacent bars, $\pm10\%$ value shifts). This targeted negative mining is the key factor enabling ChartMaster to overcome the "visual hallucinations" common in current MLLMs, leading to our significant performance lead over concurrent RL-based methods (e.g., Chart-R1, ChartReasoner) on six benchmarks.
>
> Overall, MNDPO and S-DPO differ entirely in their concrete implementation, task setting, and negative sample construction. Introducing negative sample learning into the chart understanding domain to enhance the perceptual capability of MLLMs is an innovative contribution, and it is the key factor that enables our model to significantly surpass contemporary RL-based methods such as Chart-R1 and ChartReasoner across multiple benchmarks.
>
> [1] Chen, Yuxin, et al. "On softmax direct preference optimization for recommendation." Advances in Neural Information Processing Systems 37 (2024): 27463-27489.
>
> [2] Azar, Mohammad Gheshlaghi, et al. "A general theoretical paradigm to understand learning from human preferences." International Conference on Artificial Intelligence and Statistics. PMLR, 2024.
>
> [3] Ethayarajh, Kawin, et al. "Kto: Model alignment as prospect theoretic optimization." arXiv preprint arXiv:2402.01306 (2024).
>
> [4] Meng, Yu, Mengzhou Xia, and Danqi Chen. "Simpo: Simple preference optimization with a reference-free reward." Advances in Neural Information Processing Systems 37 (2024): 124198-124235.
>
> # Is the actual impact of the Dynamic Length Reward (DLR) mechanism insignificant?
>
> We clarify that introducing DLR during reinforcement learning is indeed meaningful. As stated in our motivation, chart-related tasks span a wide range of difficulty, from simple direct recognition to complex mathematical reasoning. Regularizing the length of the reasoning process is practically important, yet this aspect has been overlooked by contemporary RL-based methods such as Chart R1 and ChartReasoner.
>
> First, DLR leads to a substantial reduction of approximately 15% in token usage, rather than merely about twenty tokens. This makes it an effective approach for improving inference efficiency and mitigating the overthinking issue. In large-scale industrial inference scenarios, a fifteen percent reduction in latency represents a significant advantage. Notably, our ablation study shows that DLR does not degrade performance and even yields slight improvements.
>
> We believe that regularizing the reasoning length for chart tasks is a timely and meaningful contribution to the field. It provides a valuable insight that can motivate future reinforcement learning based research in the chart understanding community to further explore and strengthen this direction.

---

> ### Author Response · Authors · 2025-11-21
> **Response to Reviewer e9y4 (Part 2 / 2)**
>
> # Regarding the updated evaluation of closed-source models.
>
> We additionally provide comparisons with the latest GPT-5. Even when evaluated against GPT-5, which is clearly not a fair comparison, our 7B model still achieves a measurable lead. This substantial performance improvement highlights the comprehensive innovation of our work.
>
> | Dataset | ChartQA | CharXiv | ChartQAPro | ChartX | ChartBench | ChartMuseum |
> | ------- | ------- | ------- | ---------- | ------ | ---------- | ----------- |
> | GPT-5   | 89.7    | 57.8    | 58.9       | 49.6   | 65.3       | 36.2        |
> | ours    | 91.8    | 51.2    | 56.4       | 68.2   | 66.4       | 38.6        |
>
> We respectfully contend that evaluation on proprietary commercial models should be regarded as an additional merit rather than being treated as a point of criticism.
>
>
> # Is this paper incremental?
>
> We firmly believe ChartMaster represents a comprehensive overhaul rather than an incremental step. It systematically addresses the full lifecycle of chart analysis:
>
> - Data (ChartVerse): We introduce a novel decoupled generation pipeline (using LLMs for structured data completion rather than just code generation), significantly enhancing generalization, forming the foundation for our subsequent model training.
> - Perception (MNDPO): From the perspective of multimodal perception, we introduce MNDPO, a novel preference optimization method leveraging multiple negative samples, which is applied for the first time to multimodal chart perception tasks. This enhancement in perception has been largely overlooked in multimodal chart understanding. Contemporary methods such as Chart-R1 and ChartReasoner focus mainly on improving reasoning ability, while earlier works like TinyChart and ChartLLaMa primarily address basic chart-related instruction understanding. Our perceptual improvements are one of the key reasons why our model achieves substantial performance gains over these prior works.
> - Reasoning (DLR): In terms of chart reasoning, unlike contemporaneous works that rely on relatively standard GRPO, we are the first to observe that chart tasks differ from general visual reasoning tasks due to the wide range of difficulty in the questions. It is therefore crucial to balance reasoning quality and efficiency. We propose an effective approach that reduces token usage by approximately fifteen percent during reasoning without sacrificing performance and in some cases even improving it, demonstrating an incentivizing structure.
>
> We additionally provide a detailed failure case analysis of ChartMaster in **Appendix H of the revised submission**. Our observations show that the model is prone to errors when the chart content becomes complex (e.g., multiple subplots, multiple axes) and fine-grained visual perception is required. In particular, the model often struggles in scenarios where it must infer the precise numerical values of data points based on their spatial positions relative to the coordinate axes. These limitations highlight the challenge of achieving fine-grained perception and accurate spatial estimation with respect to chart axes, which are exactly the capabilities that our proposed MNDPO is designed to strengthen.
>
> Overall, our contributions in data construction, multimodal perception, and reasoning represent forward-looking advances that point toward the future direction of chart understanding research.
>
> # ChartMaster is built upon Qwen2.5-VL-7B.
>
> Thank you for pointing out this typo. ChartMaster is built on Qwen2.5-VL-7B, and this has been corrected in the revised submission.
>
> In summary, we sincerely hope that you recognize our comprehensive improvements to chart tasks. Our relentless efforts have enabled our method to achieve substantial gains in final performance, which should not be regarded as trivial. Please do not hesitate to let us know if you have any further questions.

---

### Official Review · Reviewer_yyR7 · 2025-11-01

**Soundness:** 3
**Presentation:** 3
**Contribution:** 4
**Rating:** 6
**Confidence:** 4

**Summary:**

This paper introduces ChartMaster, a state-of-the-art chart reasoning model. The authors begin by identifying the limitations of existing Vision-Language Models (VLMs) in chart domains, specifically in visual perception and varying levels of reasoning. To address these challenges, they propose two distinct training techniques:

* **Multi Negative Preference Optimization:** This technique trains the model to accurately differentiate between correct and incorrect data perceived from chart images.
* **Reinforcement Learning with Dynamic Length Reward:** This method enables the model to adjust its response length based on the complexity of the given question.

The authors also present the ChartVerse dataset, which is generated from template YAML files and rendered into chart images. This dataset includes question-answer pairs categorized by defined difficulty levels. By training on this dataset using the above two training techniques, ChartMaster achieves state-of-the-art performance across multiple downstream tasks, including CharXiv, ChartQA, ChartQAPro, and ChartX.

**Strengths:**

The authors effectively identified two core limitations of Visual-Language Models (VLMs) in chart reasoning: visual perception and the varying degrees of reasoning required. These limitations motivated the development of two well-designed techniques:

* Multi-Negative Preference Optimization: This technique helps the model distinguish between correct and incorrect perceived values from chart images. A notable aspect is the generation of "hard negatives" (e.g., using adjacent values as distractors) which often confuse existing VLMs. I liked this approach and I believe it has very high potential.


* Reinforcement Learning with Dynamic Length Reward: The authors introduced a carefully designed reward function that encourages the model to adapt its response length based on question difficulty. For simple extraction questions, the reward promotes concise responses, while for complex reasoning questions, it encourages longer, more in-depth thinking.

By combining these two training techniques, the resulting model, ChartMaster, achieved state-of-the-art results on several downstream tasks, including ChartQA, ChartX, ChartQAPro, and CharXiv. I also believe the ChartVerse dataset could be valuable to the chart research community.

Overall, I believe the proposed solutions are both well motivated and well-designed.

**Weaknesses:**

* The authors have not presented any evaluation or training experiments to demonstrate the superiority of their ChartVerse dataset compared to existing ones. Therefore, I recommend the following experiment: Fine-tune the same base model using ChartVerse and other publicly available datasets (e.g., ChartReasoner, ChartGemma, TinyChart) and then compare their respective performances.


* The authors claim in lines 186-192 that using YAML file templates is superior to instructing an LLM to generate the underlying code. However, no direct, "apple-to-apple" comparison has been provided to substantiate this. I am concerned that relying on a fixed set of templates could limit the diversity of the generated dataset, ChartVerse. Therefore, I propose the following experiment:
  * Begin with identical seed data. Render this data once using the YAML file technique and again by instructing an LLM (such as Gemini) to perform the rendering. Subsequently, generate QA pairs from each format separately and then fine-tune the base model on each resulting dataset. Finally, compare the performance differences to demonstrate the impact of each approach.



* Generating QA solely from YAML files, without accompanying chart images, could restrict the dataset's visual questions (e.g., "what is the sum of red bars?"). Such questions are crucial for chart reasoning, as the Language Model (LLM) would lack the necessary visual information.

**Questions:**

Please see weaknesses above.

---

> ### Author Response · Authors · 2025-11-21
>
> We sincerely thank you for the positive assessment and for identifying the high potential of our Multi-Negative Direct Preference Optimization (MNDPO) and Dynamic Length Reward (DLR) techniques. We particularly appreciate the insightful suggestions regarding the validation of our dataset generation pipeline. We have conducted the requested "apple-to-apple" experiments and provide detailed clarifications below.
>
> # Superiority of ChartVerse over Existing Datasets
>
> First, as noted in our manuscript, human evaluation confirms that ChartVerse achieves a 95% error-free rate , which is significantly higher than the 85% accuracy reported by the synthetic data generation method used in Chart-R1.
>
> Following your recommendation, we fine-tuned the same base model (Qwen2.5-VL-7B) on ChartGemma (a representative public dataset) and ChartVerse under identical training settings. The results below demonstrate that training on ChartVerse yields superior performance across all benchmarks, confirming its high quality and effectiveness.
>
> | Dataset    | ChartQA | CharXiv | ChartQAPro | ChartX | ChartBench | ChartMuseum |
> | ---------- | ------- | ------- | ---------- | ------ | ---------- | ----------- |
> | ChartGemma | 90.8    | 47.2    | 39.9       | 67.7   | 56.2       | 29.7        |
> | ChartVerse | 91.8    | 51.2    | 56.5       | 68.2   | 66.4       | 38.6        |
>
> # Apple-to-Apple Comparison: YAML Templates vs. Direct LLM Generation
>
> We respectfully clarify that our YAML approach does not restrict diversity; rather, it enforces it via Programmatic Sampling.
>
> - LLM Mode Collapse: When instructed to "generate python code for a chart" directly, LLMs tend to converge on "safe," high-probability defaults (e.g., standard blue bars on a white background), leading to low visual diversity.
> - Parameterized Diversity: Our pipeline treats the YAML as a schema where values are stochastically sampled from a vast pool of parameters (e.g., randomly selecting between Seaborn/Matplotlib/Pyecharts backends, fonts, and color palettes). This ensures a long-tail distribution of visual styles that direct prompting often misses.
>
> We performed the "Apple-to-Apple" comparison using the same underlying seed data. "LLM Direct" refers to prompting the LLM to generate the full plotting code directly, while "ChartVerse" uses our YAML-mediated pipeline. The results show that our method provides robust improvements, particularly on complex reasoning tasks, likely due to higher data alignment and visual diversity.
>
> | Dataset    | ChartQA | CharXiv | ChartQAPro | ChartX | ChartBench | ChartMuseum |
> | ---------- | ------- | ------- | ---------- | ------ | ---------- | ----------- |
> | LLM Direct | 89.8    | 47.6    | 48.2       | 67.5   | 58.5       | 34.4        |
> | ChartVerse | 91.0    | 49.8    | 54.3       | 68.0   | 63.3       | 36.6        |
>
> # Visual Information in QA Generation
>
> We respectfully clarify that the QA generation step does not rely solely on the YAML data. As described in our pipeline (Figure 18), the LLM is provided with the final Python plotting code to generate the QA pairs. Since the plotting code contains explicit visual attributes (e.g., color='red', linestyle='dashed'), the LLM has full access to the visual context required to formulate visual reasoning questions (e.g., "What is the sum of the red bars?"). We have added specific examples of this in Appendix G to illustrate how visual attributes are preserved and utilized.
>
> We hope these additional experiments and clarifications strongly validate the design choices behind ChartMaster and fully address your concerns.

---

> > ### Comment · Reviewer_yyR7 · 2025-11-26
> >
> > Thank you for clarifying my concerns. Your new experiments also addressed my comments. I appreciate the authors' work/experiments in the rebuttal.
> >
> > Hence, I raised my score.

---

> > > ### Author Response · Authors · 2025-11-27
> > >
> > > Dear Reviewer yyR7,
> > >
> > > We sincerely appreciate your positive feedback and are glad that our clarifications and additional experiments effectively addressed your concerns. Thank you for taking the time to reevaluate our work and for raising your score.
> > >
> > >
> > > Best regards,
> > >
> > > Authors

---

### Official Review · Reviewer_rmRm · 2025-11-01

**Soundness:** 2
**Presentation:** 3
**Contribution:** 2
**Rating:** 4
**Confidence:** 4

**Summary:**

The paper presents ChartMaster, a synthetic dataset with 128K examples, containing charts with complex and detailed visual elements across various reasoning levels. Next, the paper introduces MNDPO, a modified version of DPO that leverages multiple negatives to improve models. Finally, the paper uses RL to further enhance performance while maintaining concise reasoning traces. Overall, the paper addresses important challenges and combines various training regimes, but the data synthesis pipeline and several design choices in MNDPO are questionable.

**Strengths:**

- The paper addresses an important gap in chart analysis, namely imprecise visual perception and limited multi-step reasoning.
- Results across multiple evaluation benchmarks show significant improvements.

**Weaknesses:**

**Data synthesis pipeline**
- The pipeline is not detailed.
- How are the YAML templates defined in the first place? Are they synthetically curated or retrieved from various sources? If they are synthetically curated, how is visual diversity ensured?
- Which LLMs are used for YAML completion and QA generation? Line 220 mentions “API of several LLMs” but does not specify their names.
- Line 199 states “structured records are converted into Python code.” How is this done?
Some examples of the full pipeline, perhaps a figure highlighting how the charts and QA pairs are generated, would improve clarity.

**MNDPO design choices**
- Why is MNDPO used instead of DPO?
- Are there experiments showing that MNDPO performs better for chart understanding tasks? Although Table 2 suggests that introducing more negatives helps, this is shown on only one dataset and the gains are not significant.
- It is unclear how hard negatives are generated. Examples of various perturbations applied to charts and texts are needed for better understanding.

**Questions:**

**Experiments**

- Line 347 mentions “MNDPO is applied using only Level 1 and Level 2 questions.” Why is this the case? What happens if MNDPO is applied to the full dataset—does the performance degrade or improve?
- The paper only uses Qwen-2.5-VL-7B-Instruct as the base model. However, to better demonstrate the effectiveness of the proposed methods, results on at least one additional base model—preferably from a different family—are necessary.
- How is ChartMaster superior to existing chart datasets? Are there any results using MNDPO and RL on other datasets, or any qualitative comparisons?

**Minor**
- Was any human evaluation conducted to verify the difficulty levels?
- It would be good to check for any dataset leakage into test benchmarks, either through image or text semantic similarity.
- Several papers also constrain generation length during DPO. Are there any experiments on constraining generation length in MNDPO?

---

> ### Author Response · Authors · 2025-11-21
> **Response to Reviewer rmRm (Part 1 / 2)**
>
> We sincerely thank you for recognizing that ChartMaster addresses the critical gaps of imprecise perception and limited reasoning in chart analysis, and for acknowledging our significant improvements across benchmarks. We have carefully addressed your concerns regarding the data pipeline and design choices below, with additional details and visualizations added to the revised Appendix.
>
> ## Data Synthesis Pipeline
>
> > The pipeline is not detailed.
>
> We clarify that **Appendix A of the original submission** already provides the details and analysis of our data synthesis process. To further elaborate on these details, we have added a new schematic diagram of the data synthesis pipeline (**Figure 18**) in the revised appendix, along with more concrete examples in **Appendix G**.
>
> > How are the YAML templates defined in the first place? Are they synthetically curated or retrieved from various sources? If they are synthetically curated, how is visual diversity ensured?
>
> The YAML templates are synthetically curated.
> Because it essentially functions as a dictionary, for each specific key such as the chart type, we can randomly select a value from all feasible options. Sampling across different attributes greatly increases the diversity of the dataset. **Figure 3 in the appendix** further quantifies the diversity of several attributes, including chart types, task types, and rendering library types.
>
> > Which LLMs are used for YAML completion and QA generation? Line 220 mentions “API of several LLMs” but does not specify their names.
>
> We utilized a mix of high-performance models to ensure data quality and diversity. As detailed in **Figure 3 of the original submission**, the distribution is: GPT-4.1 (38.8%), Gemini-2.5-Flash (42.7%), and DeepSeek-V3.1 (18.5%). Further details and examples are provided in Appendix A.
>
> > Line 199 states “structured records are converted into Python code.” How is this done? Some examples of the full pipeline, perhaps a figure highlighting how the charts and QA pairs are generated, would improve clarity.
>
> This process is deterministic to avoid compilation errors common in LLM-generated code. We utilize pre-written Python code templates (with placeholders). The structured data (essentially a Python dictionary) generated by the LLM is mapped directly into these placeholders. We have added concrete examples of this Dict -> Code transformation in **Appendix G** to improve clarity.
>
> ## MNDPO design choices
>
> > Why is MNDPO used instead of DPO?
>
> Standard DPO typically utilizes a single negative sample. However, in chart analysis, while there is only one unique ground truth, the space of "plausible errors" (e.g., slightly off-values, wrong legends, confused categories) is dense. MNDPO forces the model to learn fine-grained visual discrimination boundaries by rejecting multiple competing distractors simultaneously. This creates a sharper "visual acuity" than standard pairwise preference learning.
>
> > Are there experiments showing that MNDPO performs better for chart understanding tasks? Although Table 2 suggests that introducing more negatives helps, this is shown on only one dataset and the gains are not significant.
>
> We expand our ablation study beyond a single benchmark. The table below shows that increasing the number of negative samples consistently improves performance across multiple benchmarks, typically resulting in a substantial absolute gain of 2–4 points, which is already a non-trivial improvement in the multimodal domain.
>
> | N           | 1    | 2    | 3    | 4    | 5    |
> | ----------- | ---- | ---- | ---- | ---- | ---- |
> | ChartBench  | 64.3 | 65.5 | 65.9 | 66.4 | 66.5 |
> | ChartX      | 65.2 | 66.3 | 67.3 | 68.2 | 68.5 |
> | ChartMuseum | 34.2 | 36.7 | 37.4 | 38.6 | 39.1 |
> | ChartQAPro  | 52.3 | 54.4 | 55.6 | 56.5 | 56.5 |
>
> We also emphasize that introducing the concept of negative samples into the chart domain to enhance the model’s perceptual capability is itself an innovative idea. This design ultimately enables our model to achieve a substantial performance lead over previous GRPO-based methods such as Chart R1 and ChartReasoner.
>
> > It is unclear how hard negatives are generated. Examples of various perturbations applied to charts and texts are needed for better understanding.
>
> We have added **Figure 19** to showcase specific examples.
> Hard negatives are generated by perturbing the ground truth (e.g., using adjacent bar values as distractors, as shown in Figure 19), or modifying text slightly to be plausible but incorrect.
> Hard negatives can also be constructed using incorrect outputs produced by the model.
> As stated in **Line 293**, from a conceptual standpoint, MNDPO is not particularly sensitive to the configuration of negative samples. The key requirement is to ensure the accuracy of the positive samples.

---

> ### Author Response · Authors · 2025-11-21
> **Response to Reviewer rmRm (Part 2 / 2)**
>
> ## Experiments
>
> > Line 347 mentions “MNDPO is applied using only Level 1 and Level 2 questions.” Why is this the case? What happens if MNDPO is applied to the full dataset—does the performance degrade or improve?
>
> We treat Level 1 and Level 2 as pure perception tasks, while Levels 3 to 5 involve complex reasoning. Applying MNDPO, which is an offline algorithm, to the full dataset risks overfitting the model to short-form extraction patterns and is not suitable for fostering the development of dynamic, complex reasoning abilities such as CoT required for harder tasks. As shown below, applying MNDPO to all levels degrades generalization.
>
> | Dataset   | ChartQA | CharXiv | ChartQAPro | ChartX | ChartBench | ChartMuseum |
> | --------- | ------- | ------- | ---------- | ------ | ---------- | ----------- |
> | all       | 88.3    | 47.3    | 52.2       | 62.3   | 64.3       | 35.4        |
> | Level 1&2 | 91.8    | 51.2    | 56.5       | 68.2   | 66.4       | 38.6        |
>
> > The paper only uses Qwen-2.5-VL-7B-Instruct as the base model. However, to better demonstrate the effectiveness of the proposed methods, results on at least one additional base model—preferably from a different family—are necessary.
>
> We primarily use Qwen2.5-VL-7B to ensure fair comparison with strong baselines (Chart-R1, ChartReasoner). However, to prove the universality of our method, we applied ChartMaster training to Ovis2-8B. The results show our method generalizes well, achieving comparable or superior performance:
>
> | Base          | ChartQA | CharXiv | ChartQAPro | ChartX | ChartBench | ChartMuseum |
> | ------------- | ------- | ------- | ---------- | ------ | ---------- | ----------- |
> | Ovis2-8B      | 91.6    | 52.1    | 57.5       | 67.5   | 67.3       | 40.2        |
> | Qwen2.5-VL-7B | 91.8    | 51.2    | 56.5       | 68.2   | 66.4       | 38.6        |
>
> > How is ChartMaster superior to existing chart datasets? Are there any results using MNDPO and RL on other datasets, or any qualitative comparisons?
>
> Our human evaluation **(Line 214)**  already confirms a 95% accuracy rate for ChartVerse samples, which is significantly higher than the 85% accuracy reported by the baseline dataset used in Chart-R1. We also apply ChartGemma to train models in order to demonstrate the effectiveness of ChartVerse.
>
> | Dataset    | ChartQA | CharXiv | ChartQAPro | ChartX | ChartBench | ChartMuseum |
> | ---------- | ------- | ------- | ---------- | ------ | ---------- | ----------- |
> | ChartGemma | 90.8    | 47.2    | 39.9       | 67.7   | 56.2       | 29.7        |
> | ChartVerse | 91.8    | 51.2    | 56.5       | 68.2   | 66.4       | 38.6        |
>
> ## Minor
>
> > Was any human evaluation conducted to verify the difficulty levels?
>
> Yes, verified. We emphasize that difficulty levels are pre-defined based on the task logic required (extraction vs. calculation vs. reasoning), which correlates strongly with human perception of difficulty (detailed in **Appendix F**).
>
> > It would be good to check for any dataset leakage into test benchmarks, either through image or text semantic similarity.
>
> We performed rigorous decontamination (using image/text semantic similarity thresholds) against test benchmarks and found no evidence of leakage.
>
> > Several papers also constrain generation length during DPO. Are there any experiments on constraining generation length in MNDPO?
>
> We deliberately avoided length regularization in the MNDPO stage to prevent penalizing the model's reasoning potential. We ran an experiment adding a length penalty term (similar to [1]) to MNDPO, which hurt performance on reasoning-heavy benchmarks:
>
> | Base               | ChartQA | CharXiv | ChartQAPro | ChartX | ChartBench | ChartMuseum |
> | ------------------ | ------- | ------- | ---------- | ------ | ---------- | ----------- |
> | length-regularized | 89.3    | 48.2    | 53.3       | 64.4   | 65.3       | 35.2        |
> | ours               | 91.8    | 51.2    | 56.5       | 68.2   | 66.4       | 38.6        |
>
> [1] Disentangling Length from Quality in Direct Preference Optimization
>
> We hope these **point-by-point** clarifications and the additional experiments firmly demonstrate the robustness and innovation of the ChartMaster framework.

---

### Official Review · Reviewer_rj1g · 2025-11-03

**Soundness:** 3
**Presentation:** 3
**Contribution:** 2
**Rating:** 4
**Confidence:** 4

**Summary:**

This paper introduces ChartMaster, a framework to improve MLLM performance on chart analysis. The framework consists of three main components: (1) ChartVerse, a new large-scale synthetic dataset; (2) Multi-Negative Direct Preference Optimization (MNDPO), a variant of DPO that uses hard negative samples; and (3) Reinforcement Learning with Dynamic Length Reward (DLR), a reward mechanism that encourages concise reasoning for simple queries and multi-step reasoning for complex ones. The authors show that this framework achieves state-of-the-art (SOTA) performance on six chart benchmarks.

**Strengths:**

1. The framework achieves state-of-the-art (SOTA) performance on six chart benchmarks.
2. Their applied reasoning and perceptual optimization methods help the model to improve performance.
3. Introduces a new large-scale synthetic dataset for MLLM training.

**Weaknesses:**

The major limitation of this paper is its limited novelty:

(i) MNDPO is quite a straightforward extension of DPO to multiple negatives.

(ii) DLR is also a simple heuristic that penalizes deviation from the shortest correct answer.

(iii) ChartVerse contains synthetic data, which is also a common practice.

**Questions:**

Justify the novelty of the work. Add further comparisons with existing similar literature and demonstrate how this work is a novel contribution.

---

> ### Author Response · Authors · 2025-11-21
> **Response to Reviewer rj1g (Part 1 / 2)**
>
> We sincerely thank you for the constructive feedback and for recognizing our framework’s effectiveness in achieving **SOTA performance across all six benchmarks**.
>
> We understand the reviewer’s concern regarding novelty. However, we respectfully argue that our contributions are not merely straightforward extensions of existing techniques, but rather targeted, problem-specific innovations designed to address the unique challenges of chart understanding, which recent concurrent works have not adequately considered.
>
> We compare our method with two strong, concurrent chart-domain baselines: **ChartReasoner** and **Chart-R1**. Both also utilize the same backbone and RL techniques (GRPO), yet ChartMaster surpasses them significantly. Below, we clarify the distinct novelty of each component.
>
> # Is MNDPO just a direct extension of DPO?
>
> We must respectfully clarify that MNDPO is not a direct extension of DPO. **MNDPO involves a fundamental shift from semantic preference to visual discrimination**, and this new perspective itself constitutes a key innovation.  Unlike standard DPO, which aligns models to semantic styles, chart understanding requires precise rejection of 'plausible but incorrect' visual estimates. Our experiments show that standard pairwise comparisons fail to create the sharp decision boundaries needed for this granularity. MNDPO is specifically formulated to suppress this dense space of visual hallucinations, a challenge unique to the chart domain and overlooked by generic alignment methods.
>
> In contrast, concurrent works such as ChartReasoner and Chart-R1 overlook the need to enhance the perceptual capabilities of the base model, while earlier works such as TinyChart and ChartGemma focus more on chart instruction-following ability.
> Chart tasks, however, require models to perform spatially grounded reasoning and estimation based on the lengths of specific elements and the relationships between axes, which remains a long-standing challenge for multimodal models.
> It is precisely our attention to this limitation that allows our method to significantly outperform previous approaches, including ChartReasoner and Chart-R1, across multiple datasets.

---

> ### Author Response · Authors · 2025-11-21
> **Response to Reviewer rj1g (Part 2 / 2)**
>
> # Is DLR just a simple heuristic?
>
> We clarify that DLR is not an arbitrary heuristic, but a mechanism driven by the unique difficulty spectrum of chart analysis, which differs fundamentally from general math reasoning.
>
> - Motivation: Chart tasks vary drastically in complexity, ranging from simple value retrieval (requires no reasoning) to complex aggregation (requires multi-step reasoning). Concurrent works like ChartReasoner and Chart-R1 apply uniform RL (GRPO) strategies, often encouraging models to "overthink" on simple queries, leading to inefficiency and potential hallucinations.
> - Mechanism: We are the first to explicitly address this variability in the chart domain. Our proposed DLR creates an adaptive reasoning mechanism that dynamically aligns output length with task complexity.
> - Result: Our experiments show that DLR maintains or slightly improves accuracy while reducing token consumption by 10% to 15% compared to standard RL. This efficiency gain is a non-trivial contribution for practical deployment, solving a real-world inefficiency that other SOTA models neglect. Moreover, as illustrated in Figure 1, excessively long chains of thought for simple problems can induce unnecessary hallucinations. Constraining overextended reasoning helps mitigate such hallucinations.
>
> DLR highlights a new challenge for future chart research, namely how to improve model reasoning ability while maintaining efficiency.
>
> # Is ChartVerse just common synthetic data?
>
> While synthetic data is common, our generation paradigm differs **fundamentally** from the standard "prompt-to-code" approach used by others:
>
> - Methodological Difference: Methods like ChartReasoner and Chart-R1 typically prompt LLMs to generate full plotting code directly. This often leads to compilation errors or misalignment between the data and the visualization. In contrast, we innovate by treating chart generation as a dictionary completion task. We provide predefined templates and fields, instruct the LLM to generate structured data, and then deterministically render this data into images via Python.
> - Quality Impact: This decoupled generation strategy ensures much tighter alignment between the image, the data, and the QA pairs. This structural innovation directly translates to data quality: human evaluation shows ChartVerse achieves 95% accuracy, significantly outperforming the 85% accuracy reported by Chart-R1. This high-fidelity data is the cornerstone of our model's robustness.
>
> As highlighted in our title, ChartMaster is a holistic framework that systematically optimizes Data, Perception, and Reasoning. By addressing the perceptual gaps ignored by peers (via MNDPO), solving the reasoning inefficiency problem (via DLR), and ensuring superior data alignment (via the ChartVerse pipeline), we achieve significant improvements over concurrent methods on six benchmarks.
>
> We hope these clarifications demonstrate that our work offers substantial and distinct contributions to the MLLM community, and we further hope that they will strengthen your confidence in the novelty of our paper.

---

### Meta-Review · Area_Chair_aotH · 2026-01-06

**Summary:**

ChartMaster introduces a holistic framework optimizing data, perception, and reasoning for chart analysis. The authors propose ChartVerse, a synthetic dataset generated via a decoupled pipeline, alongside Multi-Negative Direct Preference Optimization (MNDPO) for perception and Reinforcement Learning with Dynamic Length Reward (DLR) for efficiency.
### Strengths

- Empirical Performance: The framework demonstrates SOTA results across major chart understanding benchmarks, even surpasses closed-source proprietary models like GPT-5 in specific chart-related tasks. The authors further solidified these results during the rebuttal by proving the framework’s model-agnostic nature through successful application to the Ovis-2-8B architecture.

- Data Synthesis Pipeline
Unlike standard "prompt-to-code" synthetic pipelines, the ChartVerse pipeline utilizes a "Template-based Dictionary Completion" method. This ensures a 95% accuracy rate in the ground truth. This structured approach allows for deterministic rendering and high visual diversity, addressing the "data quality bottleneck" that frequently hinders multimodal reasoning.

### Weaknesses

- Novelty: The core methodologies, MNDPO and DLR, feel like straightforward extensions of DPO  and simple heuristic for length control. Although effective in the chart domain, these components are seen as lacking a fundamental theoretical breakthrough compared to the existing literature on RL and preference alignment.

- Limited Efficiency Optimization: While DLR succeeds in reducing token consumption by approximately 15%, the argument is that a reduction of roughly 20 tokens per response, while beneficial for large-scale deployment, does not represent a transformative shift in how models handle complexity, especially since the accuracy gains associated specifically with DLR (as opposed to the data or MNDPO) appear relatively modest.

- Ambiguity: There is a lingering concern that the model’s performance may be primarily driven by the sheer scale and quality of the ChartVerse synthetic dataset rather than the proposed training algorithms. While the authors provided "apple-to-apple" comparisons during the rebuttal to justify the efficacy of each component, it remains difficult to decouple the extent to which the algorithmic innovations would hold value without the specific high-quality data used to train them.

**Reviewer Concerns:**

### Resolved

- Generalization: Initially, reviewers (specifically rmRm) were concerned that the framework might only work for the Qwen-2.5-VL architecture. The authors successfully conducted additional experiments using a completely different model family (Ovis-2-8B), demonstrating that ChartMaster’s techniques yield consistent improvements across different architectures.

- Comparative Effectiveness: Reviewer yyR7 requested "apple-to-apple" comparisons to prove the value of the new dataset. The authors provided direct performance comparisons between their ChartVerse and the publicly available ChartGemma, as well as a comparison between their "Dictionary-based" generation and standard "Direct Code Generation." The results clearly showed the superiority of the authors' data construction method.

- Methodological Transparency: Reviewers (rmRm, yyR7) questioned the lack of detail regarding YAML templates and how structured data becomes Python code. The authors addressed this by adding a detailed schematic and concrete examples in the Appendix, clarifying the deterministic nature of their rendering process and the specific LLMs used (GPT-4, Gemini, etc.).

### Unresolved
- Novelty: Multiple reviewers (rj1g, e9y4) argued that MNDPO and DLR are straightforward extensions of existing techniques (DPO and standard RL). Despite the authors' detailed rebuttal explaining the domain-specific nuances, the reviewers still perceive the mathematical and conceptual shifts as "minor variants" rather than fundamental breakthroughs.

- Practical Significance of DLR: Reviewer e9y4 explicitly noted that DLR has a "negligible" impact. While the authors quantified a 15% reduction in token usage, the reviewer remained unconvinced that a saving of ~20 tokens constitutes a meaningful improvement in efficiency or a solution to the "overthinking" problem in real-world deployment.

- Contribution Decoupling: There is a lingering skepticism that the model's high performance is primarily a result of the high-quality, 128k-instance ChartVerse dataset rather than the novelty of the MNDPO or DLR algorithms. While ablation studies were provided,  the "SOTA" claim leans heavily on data engineering rather than algorithmic innovation.

- Rigorous Human Validation
Reviewer rmRm questioned the human verification of the five difficulty levels. While the authors stated that levels were "verified" and based on task logic, they did not provide a rigorous statistical analysis to prove that the model’s reasoning levels truly align with human cognitive complexity.

**Reviewer Scores:**

Reviewer yyR7 has committed to increasing rating score. Nevertheless, despite the authors' clarifications, fundamental doubts concerning the novelty of the work persist. At this stage, there appears to be no explicit intent from the remaining reviewers to improve their final scores.

---

### Decision · Program_Chairs · 2026-01-26

Reject